# Mass-Manufacturable 3D Magnetic Force Sensor for Robotic Grasping and Slip Detection

**DOI:** 10.3390/s23063031

**Published:** 2023-03-10

**Authors:** Théo Le Signor, Nicolas Dupré, Jeroen Didden, Eugene Lomakin, Gaël Close

**Affiliations:** 1Melexis Technologies SA, CH-2022 Bevaix, Switzerland; 2Melexis Technologies NV, 3980 Tessenderlo, Belgium

**Keywords:** force and tactile sensing, magnetic sensors

## Abstract

The manipulation of delicate objects remains a key challenge in the development of industrial robotic grippers. Magnetic force sensing solutions, which provide the required sense of touch, have been demonstrated in previous work. The sensors feature a magnet embedded within a deformable elastomer, which is mounted on top of a magnetometer chip. A key drawback of these sensors lies in the manufacturing process, which relies on the manual assembly of the magnet–elastomer transducer, impacting both the repeatability of measurements across sensors and the potential for a cost-effective solution through mass-manufacturing. In this paper, a magnetic force sensor solution is presented with an optimized manufacturing process that will facilitate mass production. The elastomer–magnet transducer was fabricated using injection molding, and the assembly of the transducer unit, on top of the magnetometer chip, was achieved using semiconductor manufacturing techniques. The sensor enables robust differential 3D force sensing within a compact footprint (5 mm × 4.4 mm × 4.6 mm). The measurement repeatability of these sensors was characterized over multiple samples and 300,000 loading cycles. This paper also showcases how the 3D high-speed sensing capabilities of these sensors can enable slip detection in industrial grippers.

## 1. Introduction

The human central nervous system is responsible for simultaneously modulating grip and load forces during the manipulation of a hand-held object. When deprived of this sensory information, it becomes difficult to appropriately control grip strength and avoid object slippage [1]. Similarly, for robotic systems, vision guidance alone is not sufficient for accurate object perception and manipulation. A combination of both optical and force feedback is ideally required [2]. According to a recent review [3], there is a distinct lack of capable tactile sensors, which is limiting the development of robotic manipulation. In this work, we aimed to tackle this issue by developing a miniaturized and affordable force sensor suitable for slip detection in grippers and robotic fingers.

### 1.1. Sensor Technologies

The development of suitable tactile sensing is crucial for the future advancement of robotic control systems [4]. The most-challenging performance requirements, in terms of force, are 3D force sensing, force range, and accuracy. For skin-like applications, instead of the pure performance, the spatial resolution and the ability to cover a large area are the most important. Multiple technologies have been since proposed; see [5] for a recent review of sensors for artificial skin and the merits of each technology.

In short, piezo-resistive strain gauges excel in terms of their wide range and accuracy. MEMS technology enables compact chip-scale realization. Piezo-resistive technology is well established, and commercial solutions are readily available [6,7]. However, these commercial high-performance sensors are rigid and often limited to 1D normal force sensing. The tangential shear forces, which are critical for the stability of grasping [8,9], cannot be measured.

Three-dimensional force sensing is possible with a network of piezo-resistive (or capacitive) elements embedded in an elastomer. See [10] for a recent review. A commercial compact 3D piezo-resistive force sensor is available from Touchence [11], but the key performance metrics of this sensor (resolution and update rate) are substantially inferior to the 1D force sensors. This will be quantified in the benchmark in Section 4.

Visual-based tactile sensors, when relying on an underlying camera, offer the highest spatial density by providing an image of the contact surface. Shear and normal forces can be measured and incipient slip detected. The GelSight sensor [12] is an example of such a sensor. The key drawback of this sensor is its bulkiness (60 mm for the longest dimension [13]). This is one order of magnitude larger than the chip-scale dimension (<<10 mm) we targeted. In other optical sensors, the camera is replaced with a photo-diode, considerably reducing the size (e.g., 19 mm for the longest dimension [14]). However, this is still too large to integrate multiple sensors as an array on an end-effector or inside a robotic finger tip.

On the other hand, magnetic-based 3D force sensors are well suited for integration in a tight space while remaining economical. For example, in [15], a USD 6 magnetic force sensor was incorporated directly in the finger of a prosthetic hand. Compact chip-scale magnetic sensors measuring the 3D force vector can be fabricated, featuring a soft contact surface. This is achieved by embedding a magnet within an elastic material and mounting this on the surface of a 3D magnetometer chip [16,17,18]. Magnetic technology has been used in various incipient slip detection experiments [19,20]. More recent magnetic-based sensors incorporated biomimetic features, thereby enabling novel perception capabilities. For example, Reference [21] introduced a multimodal sensor (combining piezo-resistive and magnetic) for estimating the contact force direction, location, and joint-level torque. Reference [22] demonstrated material recognition by engineering grooves in the contact surface, reminiscent of the skin texture.

Commercial 3D magnetic-based tactile sensors, evolving from earlier research work [23], are now available from XELA Robotics under the name uSkin [24]. They can be purchased as accessories for popular cobot grippers (e.g., for the Robotiq 2F-xxx grippers, one of the best-selling gripper families for cobots [25]). The common sensor chip enabling the advances cited above [16,17,18,19,20,21,22,23] is a compact 3D magnetometer: MLX90393 [26].

In a previous work [27], we showed that the key performance limitation of this technology, namely its over-sensitivity to magnetic stray fields, could be addressed in the chip design domain, by using a multi-pixel approach operating differentially to reject stray fields. A further key issue associated with these magnetic-based force sensors is linked to the manufacturing process, specifically embedding the magnet within the soft elastomer. This is a manual process [28], which degrades repeatability and scalability, inhibiting high-volume production.

### 1.2. Preview

This paper addressed the manufacturability issue by proposing injection molding as a scalable technique for the fabrication of the elastomer containing the embedded magnet. An initial proof-of-concept series of sensor devices were fabricated and characterized in terms of sensing capability and measurement repeatability.

The paper is structured as follows: Section 2 describes the design of the prototype sensors and the manufacturing methods. Section 3 presents the experimental characterization results, in terms of mechanical and force responses. This section also showcases the sensor’s slip detection capabilities after integration into a robotic gripper. Section 4 is reserved for the discussion of the results and a state-of-the-art benchmarking summary.

## 2. Materials and Methods

### 2.1. Sensor Architecture

Figure 1 depicts the sensor concept and the actual realization side-by-side. As in the previous work [27], a single TSSOP16 package (5 × 4 mm^2^) contains two side-by-side CMOS dies. In the previous work, the system was blind to the magnetic field in the Y-direction due to the challenges of integrating multiple 3D magnetic pixels inside the same package. The present prototype includes 4 × 3D magnetic pixels in a square, whose side is 2 mm. To the best of our knowledge, this level of integration is unprecedented for a magnetic force sensor. This enables accurate and balanced sensing of the shear force in the X and Y directions, while rejecting stray fields due to the differential operation. The elastomer with an embedded disk magnet is manufactured industrially; this is the key novelty. It is then glued on the magnetometer package, using scalable semiconductor assembly processes. The repeatability of the manufacturing process is then drastically improved compared to the manual, hand-crafted mechanical transducer, as presented in previous work. This enables the production of sensors with consistent physical characteristics—a prerequisite for the cost-effective deployment in robots and grippers at a high volume.

### 2.2. Elastomer with Embedded Magnet

The magnet material is samarium cobalt (${\mathrm{Sm}}_{2}{\mathrm{Co}}_{17}$) with remanence $B_{r}\approx 1\;T$. The remanence is slightly inferior to a neodymium magnet (NdFeB with $B_{r}\approx 1.4\;T$), but the working temperature range is increased up to $350{}^{\circ }$ [29], preventing the loss of magnetization during the elastomer curing. Moreover, samarium cobalt magnets also exhibit a lower thermal drift during operation, facilitating thermal compensation. The present design is compatible with the temperature compensation, based on the on-chip temperature sensor, we introduced earlier [27].

The choice of injection molding for the elastomer manufacturing opens the design space by enabling finer geometrical shapes. We exploited this freedom to engineer a cavity, shown in Figure 2a, under the magnet to tune the overall hardness of the structure. This is another differentiator with the prior state-of-the-art, where the elastomer fills the whole available volume and hardness is solely governed by the material property. By contrast, in our design, the air cavity between the magnet and the chip increases the sensitivity of the sensor without reducing the hardness of the elastomer. A design variant with even higher sensitivity, illustrating this design freedom, will be discussed later. A one-shot injection molding process was used here for the preparation of the soft transducer with a partially overmolded magnet inside. The design ensures proper magnet fixation during processing and, subsequently, during operation against unwanted magnet displacements within the elastomer (due to external magnetic fields, for example). In the injection molding tool, the magnet is held in place by a partial nest structure along the bottom and the side of the magnet in the bottom part of the mold tool and is further fixed during mold closure by an insert pin from the top mold part. The measurement of the height of the elastomer and the height of the overmolded magnet demonstrated 18 μm and 14 μm standard deviations, respectively, as an indication of process repeatability.

Proper adhesion between the soft transducer elastomer element and the exterior mold compound of the magnetic sensor was achieved by plasma treatment before gluing. Without pre-treatment, adhesive failure was observed for a shear force of around 2.4 N (see Figure 3a). With plasma treatment, the resistance to shear strength increased more than 5-fold to above 10 N without cohesive or adhesive failures in the adhesion layer. Instead, the failure lied in the elastomer (see Figure 3b), which was ruptured by the test fixture (see Figure 3c). Throughout these tests, the magnet remained firmly in place. Note that a dome-shaped design variant, to be discussed later, was used for this test. This does not change the conclusion, as this test stresses the adhesion of the glue and the elastomer material, which are the same as in the nominal cylindrical-shaped design.

The gluing was carried out manually for these prototypes; however, all individual steps are fully compatible with automated high-volume manufacturing. The integration of different steps delivers a scalable and robust force sensor manufacturing process, where the soft transducer can be introduced as an external component in micro-electronic assembly platforms.

### 2.3. Force Calculation

A contact force, applied at the top of the sensor, deforms the elastomer, thereby displacing the magnet and modulating the magnetic field pattern. Figure 2b,c illustrate the magnetic field pattern generated by the magnet at rest and under compression. The generated magnetic field along a line in the sensor plane is plotted in Figure 2d. This shows that this structure is effective at transducing a normal compression force $F_{z}$ into a magnetic field gradient $\partial B_{x}/\partial x$. The shear forces are mostly transduced in other gradiometric components (e.g., $F_{x}\to \partial B_{z}/\partial x$). Given this relationship between the applied force and the magnetic gradient, this sensor can reject magnetic stray fields from nearby parasitic source (motors, weakly magnetized ferromagnetic objects) present in any real-world robotic environment.

Note also that the generated change of gradients are in the order of $\partial B_{x}/\partial x\approx$ 10 mT/mm.

This is well above the specified minimum operating gradient of 6 mT/mm for such a gradiometric Hall sensor [30], designed to operate in automotive conditions (*T*_max_ = 160 °C). In the commercial temperature range, the errors from the magnetic measurement chain (thermal noise, offset drifts) are expected to be in the order of 10 mN of equivalent force (0.2% of the full-scale maximum force). This is one order of magnitude below the errors due to the elastomer (hysteresis, nonlinearity, etc.). The sensor performance is, therefore, not limited by the magnetic measurement errors.

The force calculation algorithm is largely the same as in our previous work [27] (see Figure 5 and the Appendix for the full mathematical description). Briefly, the algorithm processes the 3D components of the magnetic fields sensed at the four magnetometer pixel locations. The parasitic sensitivity to temperature is first corrected by a second-order polynomial using the temperature measured by an on-chip temperature sensor. The mean magnetic field, computed by averaging the readout from the four pixels, is then subtracted from each pixel readout: $B_{i}^{\prime }=B_{i}-B^{\mathrm{mean}}$. This leaves only the differential fields with the common-mode field largely suppressed. Provided that the parasitic magnetic field sources are sufficiently far away (d>>2 mm), the parasitic fields are uniform over the sensor surface. Hence, they only introduce common-mode components. The net effect is that common-mode fields are suppressed from the measurement. The sensor is then largely immune to external magnetic fields (including the Earth’s magnetic field). The resulting magnetic features $x_{i}$, corrected for temperature and stray fields, were then used in polynomial regression models to predict the 3D force components $F_{x}=\Sigma_{i}w_{i}x_{i}$ (and the same model for $F_{y}$, $F_{y}$, but with different weights $w_{i}$). There are two notable improvements with respect to our previous work. First, the hardware now provides the complete 3D magnetic vectors at each magnetometer pixel, while the previous hardware only measured $B_{x}$ and $B_{z}$. $B_{y}$ was not measured, thereby introducing an extra error into the shear force estimation. Second, the calibration coefficients used in the algorithm are largely more repeatable. The improved repeatability decreases the calibration time substantially. This will be elaborated and quantified in Section 3.

### 2.4. Electronic Implementation

Figure 4 depicts the electronic implementation at the chip-level and PCB-level. The chip is the MLX90423 [31], a linear stray-field-immune sensor normally providing an angle output. The chip can be configured to debug mode to output digitally the 12 raw magnetic components. A standard microcontroller, on a custom-designed PCB, aggregates the raw data of the sensor. The force calculation is performed externally on a computer. The above aggregation and force calculation could be performed directly in the sensor chip, with a complete chip redesign.

## 3. Results

### 3.1. Sensor Characterization

Thanks to injection molding, different shapes of the elastomer and its cavity can be designed, thereby changing the force–displacement relationship. With the same elastomer material, we realized variants targeting different force ranges by changing the shape of the mold—everything else remaining constant. Figure 5 depicts the force–displacement relationships for our nominal design (a flat-top surface) and a design variant characterized by a dome-shaped top and thinner walls (0.6 mm instead of 0.72 mm) around the cavity. For the same total force applied at the top surface, the force transmitted to the magnet is enhanced in the variant, and the thinner walls are softer. The net effect is that the sensitivity of the variant is enhanced: a smaller force is required to displace the magnet by the same amount. Hence, the variant can resolve smaller force changes, but its full-scale force is reduced in the same proportion. The variant could be deployed in applications with a smaller force range. Conversely, by thickening the walls, variants could be designed to address a higher force range. For the rest of this paper, we focus exclusively on the nominal design with a full-scale maximum force of 5 N.

A time-lapse showing the increasing mechanical deformation of the elastomer under increasing load is shown in Figure 6. The normal force response is plotted in Figure 7a for five samples. This is shown in terms of the magnetic differential signal (difference of the planar component of two opposed sensing pixels). The repeatability of the manufacturing process translates into consistent response curves with limited part-to-part variability. These data were collected by applying a normal force on the sensor’s surface and measuring both the magnetic signals from the sensor and the normal force applied thanks to a reference load cell (ATI nano17).

To characterize the response to shear forces, the indenter was first moved down into the contact area (applied normal force), then laterally (applied shear force). Figure 7b plots the shear and normal forces applied. For a given normal force applied, there is a limit in the maximum shear force applied, and it is repeatable across samples. Indeed, increasing the lateral displacement will not change the signal as the indenter will slip on the surface of the elastomer. In terms of shear force, the sensor range is −1.5 N to 1.5 N and is naturally bipolar, unlike the normal force, which has to be compressive.

The repeatability of the sensor characteristics is a key enabler for mass production. In our previous work [27] and other prior works using a manual process for the elastomer fabrication, the part-to-part variability was so large that each sensor needed to be force calibrated on an individual basis. Each sensor needed its own set of calibration coefficients. This is illustrated in Figure 8a. It shows the normalized calibration coefficients of five sensors for the model calculating the force $F_{z}$. They vary substantially from one sensor to another. The situation was similar for the $F_{x}$ and $F_{y}$ calculation models. The root cause of this variability is the lack of dimensional control in the manual fabrication process of the elastomer with the embedded magnet and the mechanical consistency (for instance, the material non-homogeneities and voids or other defects near the surface [32]). By contrast, the industrial-grade process introduced here yields similar coefficients for all sensors (Figure 8b). Consequently, one can calibrate just one sensor of the batch and blindly reuse these coefficients on all sensors of the batch. This approach removes the force calibration bottleneck: force-calibrated samples were produced in minutes instead of hours. The obtained force responses for the 3D components are shown in the inset of Figure 8b. After such blind calibration, we obtained root-mean-squared errors of around 300 mN for the normal force and 100 mN for the shear forces. This level of error is acceptable as it is of the same order as the hysteresis error. The calibration error could be minimized in a future work by using a half-blind calibration technique [33] in which a minimalist set of critical force calibration measurements are performed individually, and these data are statistically merged with coefficient sets obtained on a subset representative of the batch.

The endurance of the elastomer was also studied by applying more than 300 k cycles of compression with a 0.65 mm indentation (corresponding to 20% of strain in the elastic material). Figure 9 plots the force measured every thousand cycles by the reference load cell. No systematic shift in the maximum or minimum forces was observed, leaving only a random cycle-to-cycle variability, largely due to the noise of the reference load cell (shown in the lower histogram, as the value measured at 0 N resulted only from the reference). This demonstrated that the mechanical properties of the elastomer do not deteriorate after 300 k cycles of compression.

To investigate the noise floor of the sensor, the distribution of reported magnetic values at rest was recorded. The effective RMS resolution (given by the standard deviation) calculated is 0.75 mN. This is approximately a four-fold improvement with respect to our prior work (2.7 mN [27]).

### 3.2. Demonstration in Gripper

A customized jaw assembly was developed to accommodate the sensors in two different robotic gripper systems. The jaw limits the compression on the sensor, keeping the deformation within the elastic regime of the material and avoiding irreversible changes. This integration can be seen in Figure 10a. The gripper used to generate the indicated results is the Schunk gripper EGI-040. Appendix A shows the integration and operation inside a Nyrio robot and gripper.

In a first experiment, the gripper grasped an object and lifted it so that it was solely carried by the gripper’s jaws via the tangential shear forces. The 3D force was then measured by both sensors integrated in the jaws. This experiment was repeated with different compression forces (corresponding to one spacing step of the jaws) and different loads. Each additional ring load had a mass of 5.54 g. Figure 10b plots the variation of the vertical force measured by the sensor during these different experimental runs. The dashed lines correspond to the actual weight of the object carried by the gripper, which is based on the number of rings. As expected, when the normal force applied was high enough (e.g., 1.4 N), the shear force measured by the two sensors increased linearly with weight, following the dashed lines. However, when the gripping strength was lower, we observed a decrease in the measured shear force, which indicated a bad grip and a risk of slippage. This threshold was observed at various levels of compression depending on the mass of the object held. This was a result of the friction limit that we highlighted earlier in Figure 7b.

Dynamic slip detection was investigated using the same experimental setup. The magnetic sensor was set in its fastest acquisition mode, enabling sampling of the raw magnetic fields at 5 kHz. The sensor output protocol was switched from serial digital to analog, thereby circumventing the timing limitations associated with serial communication. Recall that this chip is normally an angle sensor, which was operated here in slow test mode. We also skipped the force calculation temporarily to focus on the dynamics instead and explore the benefits of high-speed (5 kHz) magnetic sensing. Such a speed is readily reachable with a chip redesign. Figure 10c shows the raw signal waveform acquired and a band-pass-filtered version (between 20 and 300 Hz). At time *t* = 0 s, an object was held by the gripper. During the experiment, it was released by the gripper at a constant rate. In the first stage (yellow area in the plot), the magnetic signal slowly changed as the magnet was progressively moving away from the sensor. After some time, the grip was not strong enough to prevent the object from slipping. Small oscillations corresponding to the vibration of the object were measured by the magnetic sensor. Between the incipient slip (covered by the red area) and the drop of the object, there was a 400 ms window during which the oscillations were clearly detectable. The green area shows the relaxation of the elastomer and, thus, of the magnetic signal once the object has dropped. This experiment demonstrated the potential of this technology to prevent objects from being dropped. By monitoring these oscillations reported by the sensor, the robot would have enough time to adjust the grasp by tightening its end-effector. Similar closed-loop control experiments will be the focus of future work.

## 4. Discussion

Table 1 benchmarks the performance of the sensor presented in this work, with existing state-of-the-art force sensors. Unlike other magnetic sensor prototypes developed [18,28], the sensor presented here is immune to magnetic stray fields owing to the differential operation enabled by the multi-magnetic pixel concept. This is significant as stray fields are pervasive in a robotics environment due to the presence of nearby motors and ferromagnetic objects. A further key differentiator is the scalable manufacturing process and the resulting repeatability. When compared to commercially available optical sensors [34], this work’s magnetic sensor is significantly smaller, allowing easier integration into a variety of applications. Honeywell’s FMA force sensor, a MEMS-based piezo-resistive sensor [7], sets the benchmark in terms of repeatability, but only provides the 1D normal force component. While this is adequate for many industrial applications, it falls short of the needs of grippers in robotics. Touchence’s Shokac Chip T08 [11] provides 3D force sensing in a compact form factor, albeit at a slower output rate (100 Hz vs. 1 kHz). The specified resolution is 40 mN (compared to the 0.75 mN achieved here), highlighting the challenge of balancing the dynamic performance metrics and the 3D sensing capability. For both piezo-resistive devices [7,11] in the benchmark, the (compensated) maximum temperature is limited to 50 ${}^{\circ }C$, a general concern for piezo-resistive sensors raised in the review [10]. Despite this concern, piezo-resistive sensors constitute a solid alternative, and no single sensor (or sensing technologies) stands out. An optimum tactile sensor might be multi-modal, as suggested in [35].

This work’s force sensor advances the state-of-the-art in the manufacturability of stray-field-immune magnetic soft sensors thanks to the following three improvements. First, the industrial manufacturing process yields reliable sensors with the ability to tolerate overload forces. Second, the repeatability of the elastomer was also improved thanks to the dimensional control (with *σ* < 20 μm), enabling blind calibration of the force response, thereby removing the bottleneck of extensive individual force calibration of each sensor. Third, a family of product variants targeting different force ranges can be realized with the same elastomer, by design adjustments to the cavity dimension in the order of 100 μm. To demonstrate mass-manufacturability, we fabricated 300 elastomer parts with embedded magnets. We assembled 50 of those on top of magnetometer chips to produce complete sensors. We also showed that the calibration coefficients can be blindly copied to instances of the same production batch. Overall, our pre-series production process paves the way toward high-volume manufacturing. The sensor performance approaches the performance of rigid piezo-resistive 1D sensors. The sensor remains compact, soft, and 3D-capable—the hallmark features of magnetic force sensors that have fueled the recent advances in robotic tactile sensing in applications such as super-resolution tactile skins [18] and material classification based on texture perception [22]. Overall, this work opens the potential for mass-manufacturable compact 3D magnetic force sensors. This could enable the deployment of force sensing and slip detection in smart grippers in high-volume robotic applications.

## Figures and Tables

**Figure 1 sensors-23-03031-f001:**
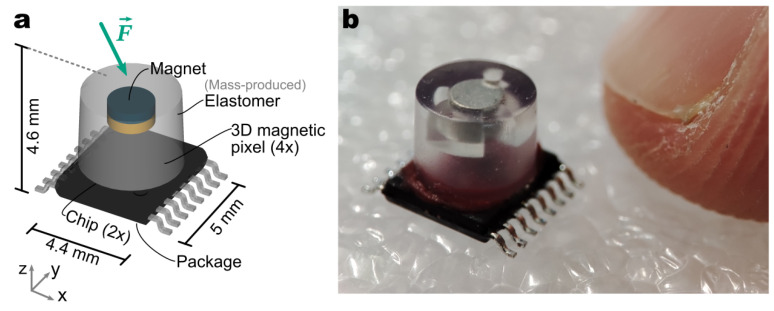
(**a**) Concept diagram of the magnetic force sensor. It consists of two magnetometer sensor chips in the same package, providing in total 4 3D magnetic pixels. A mass-produced elastomer with an embedded magnet is glued on top. The full assembly (except the leads) fits inside a cube, whose edge is 5 mm long. (**b**) Picture of the actual realization.

**Figure 2 sensors-23-03031-f002:**
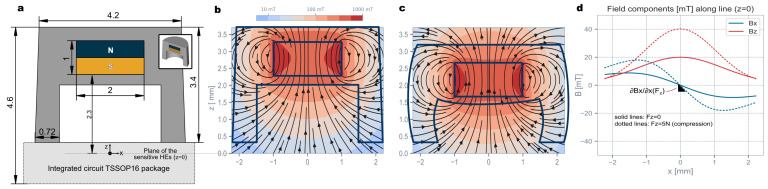
(**a**) XZ cross-section of the elastomer with the embedded magnet above the air cavity. Note that the sensor is centered on the origin, and the sensitive Hall elements (HEs) are on the z=0 plane. (**b**,**c**) Magnetic field lines at rest and under compression ($F_{z}\approx 5\;N$, based on the design estimation). (**d**) Magnetic field components along the *X* axis.

**Figure 3 sensors-23-03031-f003:**
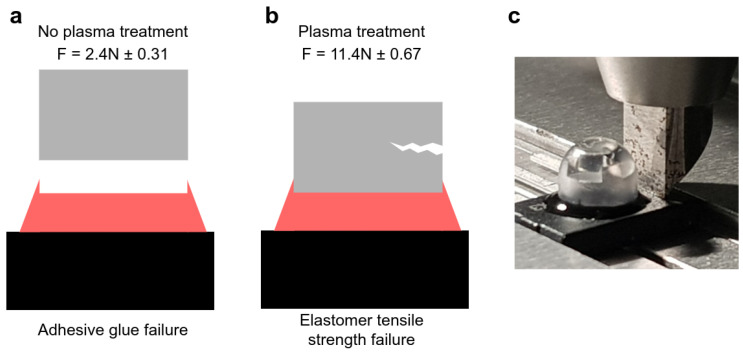
Shear strength testing of the elastomer with the embedded magnet. (**a**) When the glue is applied without plasma treatment, adhesive glue failure occurs. (**b**) Plasma treatment improves the resistance 5-fold, and the failure lies in the elastomer instead. (**c**) Picture of the testing setup.

**Figure 4 sensors-23-03031-f004:**
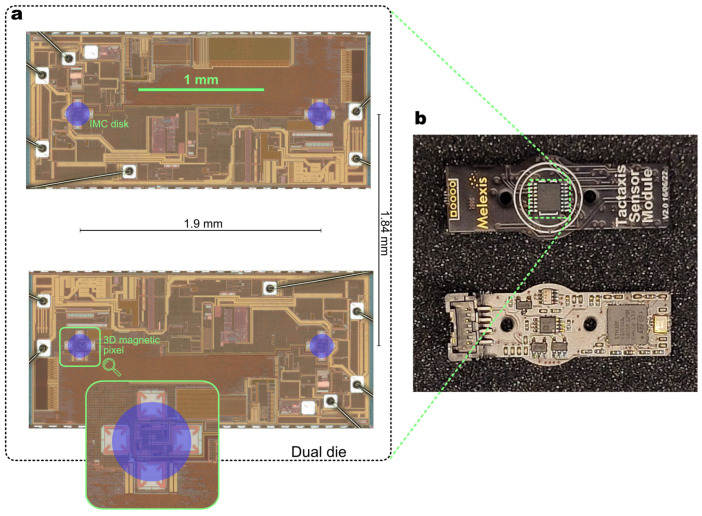
Electronic implementation. (**a**) Sensor chip micrograph with the inset zooming into one of the four 3D magnetic pixels. (**b**) Companion PCB hosting the sensor on the top face and standard components on the other face to provide a standard I2C serial interface.

**Figure 5 sensors-23-03031-f005:**
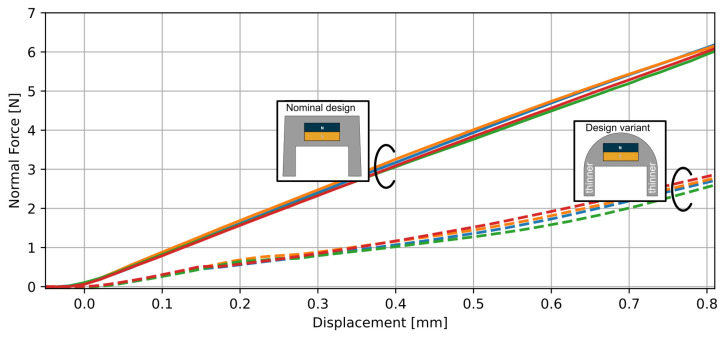
Force–displacement relationship for the nominal design and a design variant illustrating the different force ranges. 4 parts were tested in each case. Each part corresponds to a different color.

**Figure 6 sensors-23-03031-f006:**
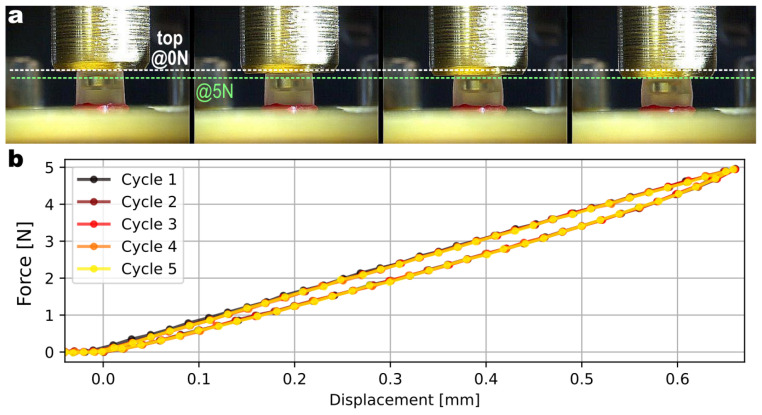
Mechanical response. (**a**) Time-lapse with increasing indentation and compression force, illustrating the mechanical deformation of the elastomer. (**b**) Applied compression force (measured by a reference load cell) as a function of the displacement with respect to the contact point over 5 cycles.

**Figure 7 sensors-23-03031-f007:**
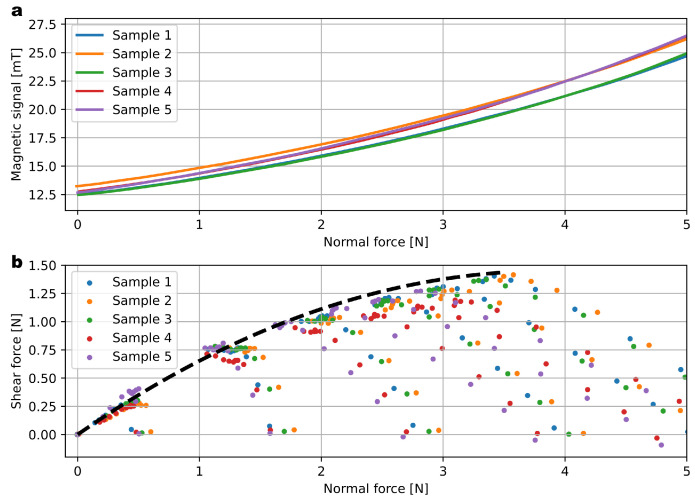
Response to applied forces. (**a**) Magnetic response to normal applied force for 5 samples, demonstrating the achieved consistency among the fabricated sensors. (**b**) Scatter plot of the shear force vs. normal force applied, demonstrating the consistency of the friction limit.

**Figure 8 sensors-23-03031-f008:**
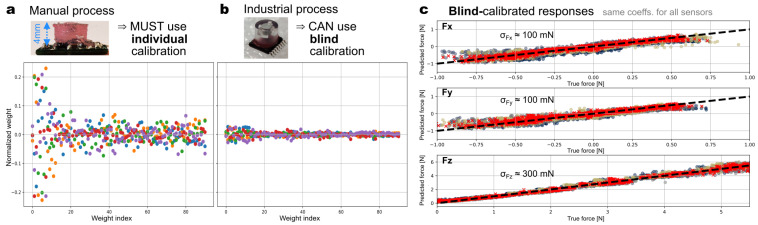
Normalized calibration coefficients (mean normalized weights: $w_{i}-\bar{w_{i}}$) of the $F_{z}$ calculation model $F_{z}=\Sigma_{i}w_{i}x_{i}$ for 5 sensors (each sensor corresponds to a different color). (**a**) Sensors with the elastomer fabricated manually (prior work), resulting in substantial part-to-part variability. (**b**) Sensors with the elastomer fabricated industrially (this work). The dispersion is under control. (**c**) Force responses (predicted versus ground truth) of 5 sensors with the elastomer fabricated industrially and blindly calibrated (same coefficients $w_{i}$ used for all sensors).

**Figure 9 sensors-23-03031-f009:**
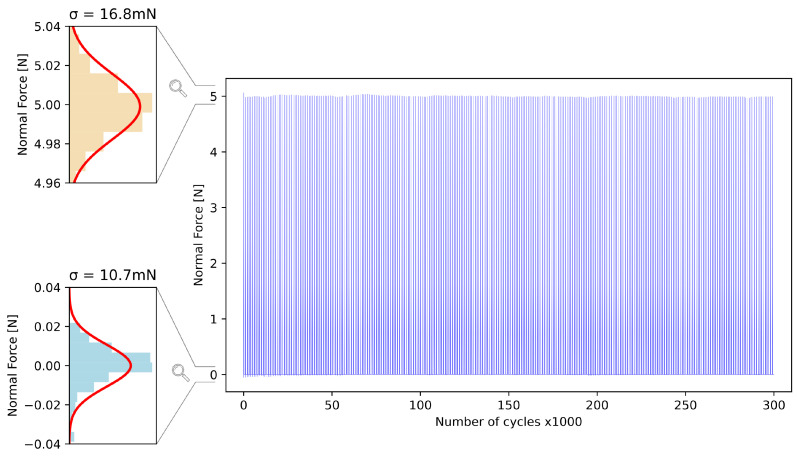
Repeatability error when cycling the normal force between 0 and 5 N up to 300,000 cycles.

**Figure 10 sensors-23-03031-f010:**
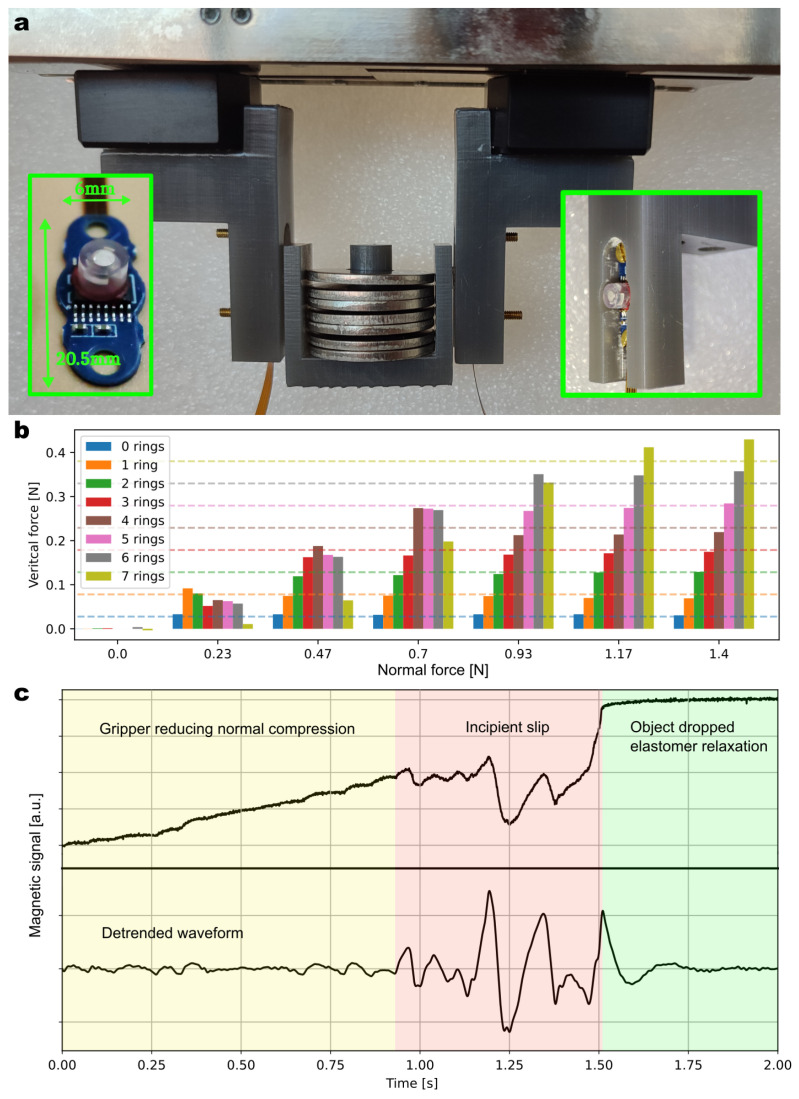
(**a**) Picture of the experimental setup, where two sensors are integrated in the jaws of a Schunk gripper. (**b**) Shear force measurement at different compression forces and mass loads. (**c**) Dynamic waveform acquired at the time of slippage: raw waveform (**top**) and detrended waveform (**bottom**).

**Table 1 sensors-23-03031-t001:** Benchmark against other force sensors.

	This Work	Magnetic Prototypes			Products	
		Sc. Robotics’21 [18]	ReSkin/CoRL’22 [28]	Contactile [34]	Honeywell [7]	Touchence [11]
Technology	Magnetic	Magnetic	Magnetic	Optical	Piezo-resistive	Piezo-resistive
Stray field immunity	✓Yes	No	No	✓Yes	✓Yes	✓Yes
Sensed quantity	3D force	3D force	3D force	3D force	1D force	3D force
Mass-manufacturable	✓Yes	-	No	✓Yes	✓Yes	✓Yes
Repeatability error ^1^ (conditions)	*σ* = 1% (300 kcycles @5N)	*μ* = 4% (30 kcycles @1N)	Not quantified (50k interactions)	-	*μ* = 1.4% (1 Mcycles @5N)	Not quantified (1 Mcycles)
Update rate	50 Hz ^2^	10 Hz	2 kHz	-	>1 kHz	100 Hz
Longest dimension	5 mm	5 mm	8 mm	19 mm	5 mm	7.4 mm

^1^ Repeatability error might display a systematic shift (*μ*) due to permanent change, in addition to the random part (*σ*). ^2^ Limited by the serial protocol in this prototype; 1 kHz reachable without loss of performance with a chip redesign.

## Data Availability

The data presented in this study are available upon request from the corresponding author.

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
