# Peer review of "Mass-Manufacturable 3D Magnetic Force Sensor for Robotic Grasping and Slip Detection"

_sensors, 2023, doi:10.3390/s23063031_

Round 1
Reviewer 1 Report
The paper presents a 3D magnetic force sensor prototype suitable for mass manufacturing. The authors conducted comprehensive experiments to evaluate the sensors' characteristics and successfully applied the sensor in a slip-detection task with an industrial gripper. The structure of the article is clear, and the experiments are comprehensive. But the writing language needs to be improved, and some arguments must be clarified.
- Abstract, L1-2, this claim is powerful but not accurate. There are several industrial grippers and industrial-grade 3D force sensors on the market. For example, the grippers from Robotiq have available tactile sensors for a long time. I suggest the authors rewrite the sentences or provide more references to support the claim.
- L34-35, the statement that MEMS sensors are "limited to 1D normal force sensing" is false. Please check the Shokac Chip sensor by Touchence Inc. This sensor was developed around five years ago.
- L137-138: the argument for this sentence is unclear; please clarify.
- The authors refer to their previously published paper [21] several times; hence have omitted some important information in the current paper. For example, in L139-140, the regression algorithm is crucial for the readers to understand the force calculation. However, it is not presented here. I suggest the authors add all the necessary information to make the current paper more coherent and easy to understand.
- L142-145, it is not clear why the sensor is largely immune to external magnetic fields in this case; please clarify.
- Fig. 4, please increase the figure size. The annotation on the picture is not clear.
- L183: if the authors were to take the previous work as a baseline and make a comparison, please provide accurate reference data.
- L184-185: it is unclear how the air cavity and the new design freedom offer improved trade-off results.
- L25: the title of subsection "1.1 Context and Goal" seems redundant.
- L43: "for the present applications," it is unclear which applications the authors refer to, as no specific applications have been mentioned previously.
- L73: "state of the art" should be "state-of-the-art," a single word with hyphens. Same for L246.
- The title of this paper mentions "mass-manufacturable"; however, this property has not been related to the properties of the sensor. In other words, which features of the sensor make them "mass" manufacturable? This needs to be highlighted in the text.
- The language needs to be improved to make the paper more fluent.
Author Response
Thank you for your constructive comments and bringing the Shokac Chip by Touchence to our attention. All your comments have been addressed in the revised manuscript. See the point-by-point answers below.
- "The claim (Industrial grippers today lack the sense of touch, preventing the manipulation of delicate 1objects. A cost-effective industrial-grade 3D force sensor is missing) is powerful but not accurate. There are several industrial grippers and industrial-grade 3D force sensors on the market. For example, the grippers from Robotiq have available tactile sensors for a long time. I suggest the authors rewrite the sentences or provide more references to support the claim".
- CHANGED: we included in the introduction a discussion of the force sensors available commercially as accessories for the popular Robotiq grippers from XELA robotics. We also reformulated the abstract with a more moderate claim.
- NOTES: Robotiq only commercialize a force sensor for the joint, but not a tactile sensor for measuring the contact force. They had a tactile sensor in beta testing in 2016 [src], but this did not evolve into a product.
- The statement that MEMS sensors are "limited to 1D normal force sensing" is false. Please check the Shokac Chip sensor by Touchence Inc. This sensor was developed around five years ago.
- CHANGED: The Schokac chip is now cited and included in the benchmark table as a commercial solution. We also cite a review [Yeh, S.K. 2021, CMOS-Based Tactile Force Sensor: A Review, IEEE Sens. J.] discussing pother MEMS 3D force sensors.
- NOTES: We were not aware of this chip. Thanks for putting it on our radar.
- The argument for this sentence ("The sensor performances 137
are then not limited by the magnetic errors") is unclear; please clarify.- CHANGED: We added a paragraph explaining and quantifying this reasoning. In short, the underlying magnetic sensor that we used (we added the datasheet in references) is designed to operate in the automotive temperature range with even lower field. The errors introduced by the magnetic measurement chain are much lower than the errors of the elastomer.
- CHANGED: We added a paragraph explaining and quantifying this reasoning. In short, the underlying magnetic sensor that we used (we added the datasheet in references) is designed to operate in the automotive temperature range with even lower field. The errors introduced by the magnetic measurement chain are much lower than the errors of the elastomer.
- The authors refer to their previously published paper [21] several times; hence have omitted some important information in the current paper. For example, in L139-140, the regression algorithm is crucial for the readers to understand the force calculation. However, it is not presented here. I suggest the authors add all the necessary information to make the current paper more coherent and easy to understand.
- NOTES. The algorithm is fully described in our previous work graphically with a block diagram and the math in appendix.
- CHANGED: We included a better citation and a standalone description of the algorithm in one new paragraph.
- "The force calculation algorithm is largely the same as in our previous work [27 , Fig. 5 and appendix for the full mathematical description]. Briefly, the algorithm processes..."
- "The force calculation algorithm is largely the same as in our previous work [27 , Fig. 5 and appendix for the full mathematical description]. Briefly, the algorithm processes..."
- It is not clear why the sensor is largely immune to external magnetic fields in this case; please clarify.
- CHANGED: the stray-field rejection step of the algorithm is now described. It is incorporated in the brief description of the algorithm from the previous point.
- CHANGED: the stray-field rejection step of the algorithm is now described. It is incorporated in the brief description of the algorithm from the previous point.
- Fig. 4, please increase the figure size. The annotation on the picture is not clear.
- CHANGED as suggested. We also increased the size of a few other figures.
- CHANGED as suggested. We also increased the size of a few other figures.
- If the authors were to take the previous work as a baseline ("This is about a four-fold improvement with respect to our prior work") and make a comparison, please provide accurate reference data
- CHANGED: we added the explicit noise of the previous work.
- CHANGED: we added the explicit noise of the previous work.
- It is unclear how the air cavity and the new design freedom offer improved trade-off results.
- CHANGED: we added a paragraph describing a design variant exploiting the design freedom. We also added an experimental figure (Fig 5) to illustrate the different force range and different sensitivity of the variant.
- CHANGED: we added a paragraph describing a design variant exploiting the design freedom. We also added an experimental figure (Fig 5) to illustrate the different force range and different sensitivity of the variant.
- The title of subsection "1.1 Context and Goal" seems redundant.
- CHANGED: we removed this subsection heading.
- CHANGED: we removed this subsection heading.
- "For the present applications," it is unclear which applications the authors refer to, as no specific applications have been mentioned previously.
- CHANGED: the applications are now explicitly mentioned.
- CHANGED: the applications are now explicitly mentioned.
- "state of the art" should be "state-of-the-art," a single word with hyphens.
- CHANGED: we added the hyphen when "state-of-the-art" is used as an adjective at 2 locations.
- NOTES: There is one location where it is used as a noun. We didn't add the hyphens there, following this guidance: https://grammarist.com/usage/state-of-the-art/.
- The title of this paper mentions "mass-manufacturable"; however, this property has not been related to the properties of the sensor. In other words, which features of the sensor make them "mass" manufacturable? This needs to be highlighted in the text.
- NOTES: Based on the reviews, we realized we had not emphasized enough the significance of the sensor repeatability from a mass-manufacturing perspective,
- CHANGED: we added a long paragraph and a figure (Fig. 8) explaining how the repeatability enables blind calibration, reducing the calibration time from hours to minutes. We also added in the conclusion the 3 key points that support our claim for mass manufacturability. We also clarified the number of sensors we built (n=300) to demonstrate our pre-series fabrication.
- The language needs to be improved to make the paper more fluent
- CHANGED: the text was reviewed and edited by a native speaker colleague.
Reviewer 2 Report
In this paper, a prototype suitable for mass production, an affordable industrial-grade 3D force sensor, is presented. A sensor with repeatability and scalability of measurements to high volume production. The prototype contains two integrated circuits, providing total four magnetic 3D pixels, longest dimension is 5mm.
The authors argue that they open the door for scalable technology and force sensing for high-volume robotic applications.
The work is very applicative and less scientific while retaining its valid specific applications in the industrial field and from the tests performed to test the sensor it should be a good industrial application and when no criticism at work
The treatment is good with logic and clarity of exposition. The written English should be improved a bit and also the citations should be strengthened.
It is suggested to add the following:
- Cammarata, A., Maddìo, P. D., Sinatra, R., Rossi, A., & Belfiore, N. P. (2022). Dynamic model of a conjugate-surface flexure hinge considering impacts between cylinders. Micromachines, 13(6), 957. https://doi.org/10.3390/mi13060957
Author Response
Thank you for your comments, we addressed them as follows.
- The written English should be improved a bit.
- CHANGED. The manuscript was reviewed and edited by a native speaker.
- CHANGED. The manuscript was reviewed and edited by a native speaker.
- The citations should be strengthened.
- CHANGED. We added the 8 references based on the comments of all 3 reviewers.
- [Touchence, “Shokac Chip 6DoF-C18 Datasheet.” Available: http://www.touchence.jp/en/products/chip01.html.]
This device was suggested by a reviewer as a competitive 3D MEMS-based force sensor. It is now integrated in the benchmark table. - [Xela Robotics, “Integrations by XELA Robotics.” Available: https://xelarobotics.com/en/integrations.]
XELA Robotiscs offers a commercial tactile 3D sensor for industrial grippers. This is added in the introduction. - [A. Votta et al “Force-Sensitive Prosthetic Hand with 3-axis Magnetic Force Sensors,” IEEE Intl. Conf. on Cyborg and Bionic Systems (CBS), 2019.]
This reference illustrates one of the application use cases and highlights the compactness and cost-effectiveness of magnetic force sensors. - [M. Berger et al, “Half-Blind Calibration for the Efficient Compensation of Parasitic Cross-Sensitivities in Nonlinear Multisensor Systems,” IEEE Sens. J., 2019.]
Our wok uses a blind calibration technique, that we emphasized in the revised manuscript, as this is a key enabler for manufacturing. However, this approach is limited, and the above reference provides some outlook for future work. - [N. Dupré et al, “A stray-field-immune magnetic displacement sensor with 1% accuracy,” IEEE Sens. J., 2020].
This is one of our previous works which provides a baseline for the magnetic performance (without the elastomer). - [S.-K. Yeh et al “CMOS-Based Tactile Force Sensor: A Review,” IEEE Sens. J., 2021]
This review covers several compact integrated MEMS 1D and 3D force sensors. This has been added in the introduction. - [W. A. Friedl and M. A. Roa, “Experimental Evaluation of Tactile Sensors for Compliant Robotic Hands,” Front Robot AI, 2021].
This reference provides an experimental benchmark of various tactile sensors, with the best results obtained with multi-modal approach (magnetic force sensor and IMU). This is cited in the conclusion.
- [Touchence, “Shokac Chip 6DoF-C18 Datasheet.” Available: http://www.touchence.jp/en/products/chip01.html.]
- CHANGED. We added the 8 references based on the comments of all 3 reviewers.
- It is suggested to add the following: [Cammarata, A., Maddìo, P. D., Sinatra, R., Rossi, A., & Belfiore, N. P. (2022). Dynamic model of a conjugate-surface flexure hinge considering impacts between cylinders. Micromachines, 13(6),957. https://doi.org/10.3390/mi13060957]
- NOTES. This reference describes a MEMS-based flexure hinge. We felt that it was more relevant to cite complete MEMS force sensors (like the Shokac Chip and the review by [S.-K. Yeh] from the point above). We didn't include the suggested reference.
Reviewer 3 Report
The paper discussed the performance of the 3D magnetic force sensors in terms of manufacturability and measurement repeatability. A novel magnet elastomer prototype fabricated with injection molding was presented in this paper., which addressed the repeatability problem of the prior hand-crafted mechanical transducer. Differential operation of the data from four 3D magnetic sensors enabled the system immune to magnetic stray fields. The paper also fits the novel sensor in an industrial gripper to illustrate its characteristics.
While the paper is well organized and the description is clear, it is weak in the performance test of the whole sensing system. The paper has taken the gripper experiment and the slip detection, but there is no standard reference of the force in that occasion to prove the measurement accuracy. Below are my detailed comments:
- There is no specific description of the logic and implementing path of the force calculation process: although the paper has simulated the magnetic distribution with the normal compression to the elastomer, the real situation is more complicated. We can’t agree that the sensor’s result is reasonable just with the limited content in the paper.
- The paper lacks some important performance tests of the novel sensor: with the goal of mass manufacturability, we should not only focus on the simple compression repeatability test. More performance descriptions should be discussed such as the test accuracy.
- The repeatability of the sensor response curves should have a more detailed discussion: another advantage of the sensor’s structure mentioned in the paper is the sensitivity to change the elastomer’s hardness by designing the air cavity. Two sensor response curve test sets of different hardness elastomers can be more persuasive of the repeatability of the sensor response curves.
- While the author has referred to plenty of previous work, I think some of the articles could be helpful in this subject
[1] Wang, Guangqing, et al. "Nonlinear magnetic force and dynamic characteristics of a tri-stable piezoelectric energy harvester." Nonlinear Dynamics 97 (2019): 2371-2397.
[2] Zhang, Yingtao, et al. "The Design of Reflected Laser Intensity Testing System and Application of Quality Inspection for Laser Cladding Process." Machines 10.10 (2022): 821.
[3] Votta, Ann Marie, et al. "Force-sensitive prosthetic hand with 3-axis magnetic force sensors." 2019 IEEE International Conference on Cyborg and Bionic Systems (CBS). IEEE, 2019.
Author Response
Thank you for your constructive comments, which made us realize we needed to provide more test results. We have added two results figures. Below is our
point-by-point response to each comment.
- There is no specific description of the logic and implementing path of the force calculation process.
- NOTES. The logic and the algorithm are fully described in our previous work: there is a block diagram, and the mathematical description of the algorithm in an appendix.
- CHANGED. We provided a clearer citation, and included a standalone description in a long paragraph:
"The force calculation algorithm is largely the same as in our previous work [27 , Fig. 5 156 and appendix for the full mathematical description]. Briefly, the algorithm processes ..."
- The paper lacks some important performance tests of the novel sensor with the goal of mass manufacturability, we should not only focus on the simple compression repeatability test. More performance descriptions should be discussed, such as the test accuracy.
- CHANGED. We added in the inset of Fig. 8 the results of an accuracy test for the three force components, after a "blind" calibration. The obtained root-mean-squared errors (with respect to a reference load cell) are also discussed in the text.
- NOTES. The blind calibration is a key enabler for mass manufacturing as it enables to reuse reference coefficient sets across a batch of sensor, instead of individual calibration of each sensor.
- The repeatability of the sensor response curves should have a more detailed discussion: another advantage of the sensor’s structure mentioned in the paper is the sensitivity to change the elastomer’s hardness by designing the air cavity. Two sensor response curve test sets of different hardness elastomers can be more persuasive of the repeatability of the sensor response curves.
- CHANGED. We added a design variant and characterized its force-displacement response in Fig. 5. This variant uses a different shape (especially the cavity wall thickness) with the same elastomer material. It achieves a 3x higher sensitivity (but a reduced range in the same proportion). The shape is different.
- NOTES. We didn't include another elastomer material. Our research, so far, has focused on a single material to dig deep into the sensor characteristics. Furthermore, as proven by the design variant, we can still change the force range by a substantial factor from our nominal design with just shape variations. We plan to investigate other materials at a later stage. This is beyond the scope of this revision and its 10-day deadline.
- While the author has referred to plenty of previous work, I think some articles could be helpful in this subject: [1] Wang, Guangqing, et al. "Nonlinear magnetic force and dynamic characteristics of a tri-stable piezoelectric energy harvester." Nonlinear Dynamics 97 (2019): 2371-2397. [2] Zhang, Yingtao, et al. "The Design of Reflected Laser Intensity Testing System and Application of Quality Inspection for Laser Cladding Process." Machines 10.10 (2022): 821. [3] Votta, Ann Marie, et al. "Force-sensitive prosthetic hand with 3-axis magnetic force sensors." 2019 IEEE International Conference on Cyborg and Bionic Systems (CBS). IEEE, 2019.
- CHANGED. [3] was included to illustrate the compactness and cost-effectiveness of 3D magnetic force sensor in robotic fingers. 7 other references were added. In particular, we added two references discussing other compact 3D force sensors, based on MEMS technologies. A commercial product, and a recent review.
- Touchence, “Shokac Chip 6DoF-C18 Datasheet.” Available: http://www.touchence.jp/en/products/chip01.html.
- S.-K. Yeh, M.-L. Hsieh, and W. Fang, “CMOS-Based Tactile Force Sensor: A Review,” IEEE Sens. J., Jun-2021.
- NOTES. We did consider [1] and [2] but didn't see a strong relationship with our work. Ref [1] could be interesting, should we consider a MEMS-based spring system instead of the elastomer.
- CHANGED. [3] was included to illustrate the compactness and cost-effectiveness of 3D magnetic force sensor in robotic fingers. 7 other references were added. In particular, we added two references discussing other compact 3D force sensors, based on MEMS technologies. A commercial product, and a recent review.
Round 2
Reviewer 1 Report
The authors have well addressed all comments raised in the previous review. I recommend publishing the manuscript in its present form.